# Improving Parallel Program Performance with LLM Optimizers via Agent-System Interfaces

Anjiang Wei [* 1]  Allen Nie [* 1]  Thiago S. F. X. Teixeira [2]  Rohan Yadav [1]  Wonchan Lee [3]  Ke Wang [4]  Alex Aiken [1]

## Abstract

Modern scientific discovery increasingly relies on high-performance computing for complex modeling and simulation. A key challenge in improving parallel program performance is efficiently mapping tasks to processors and data to memory, a process dictated by intricate, low-level system code known as *mappers*. Developing high-performance mappers demands days of manual tuning, posing a significant barrier for domain scientists without systems expertise. We introduce a framework that automates mapper development with generative optimization, leveraging richer feedback beyond scalar performance metrics. Our approach features the Agent-System Interface, which includes a Domain-Specific Language (DSL) to abstract away the low-level complexity of system code and define a structured search space, as well as AutoGuide, a mechanism that interprets raw execution output into actionable feedback. Unlike traditional reinforcement learning methods such as OpenTuner, which rely solely on scalar feedback, our method finds superior mappers in far fewer iterations. With just 10 iterations, it outperforms OpenTuner even after 1000 iterations, achieving $3.8\times$ faster performance. Our approach finds mappers that surpass expert-written mappers by up to $1.34\times$ speedup across nine benchmarks while reducing tuning time from days to minutes.

## 1. Introduction

Modern scientific discovery depends on advanced software tools for modeling and simulation (Stocks et al., 2024; Wang et al., 2024; Ltaief et al., 2024). Computational scientists, including physicists, chemists, and biologists, rely on high-performance computing to tackle complex problems. These scientific computations dominate workloads on the world's most powerful supercomputers (Exascale Computing). However, many domain scientists lack expertise in computer science, and therefore having difficulties in optimizing their programs because of the complexity and scale of the underlying machines. Even for experts, finding and fixing performance problems resulting from program modifications or when porting to a new machine is often time-consuming. Any progress on automating performance tuning is of great benefit in this domain.

Task-based programming (Slaughter et al., 2015; Bauer et al., 2012; Augonnet et al., 2009; Chamberlain et al., 2007; Moritz et al., 2018; Barham et al., 2022) has emerged as a promising approach to high performance computing. The paradigm involves decomposing computations into independent *tasks* that communicate exclusively through their arguments. A key advantage of task-based systems is that the performance tuning problem is factored out into a separate *mapping*: an assignment of tasks to processors and data to particular memories. High-quality mapping, achieved through a well-designed *mapper* (implemented as code), can significantly improve performance, often by an order of magnitude (Galvez et al., 2017).

However, currently writing mappers remains a labor-intensive process, as it requires deep knowledge of applications, hardware, and low-level system APIs. In addition, this process is highly application-specific, input-specific, and machine-specific, often taking experts several days of meticulous tuning to achieve high performance. This challenge is especially pronounced for domain scientists, who typically lack the necessary expertise in computer systems and code optimization. Automating mapper development would enable scientists to focus on their own domain of expertise while fully utilizing the capabilities of high-performance computing systems.

In this paper, we introduce a system powered by large language models (LLMs) to **automate both the generation and optimization of mapper code**. The first challenge stems from the *complexity of generating mapper code* due to the original low-level programming system, which ex-

---
[*]Equal contribution  [1]Stanford University  [2]Intel  [3]NVIDIA  [4]Nanjing University. Correspondence to: Anjiang Wei <anjiang@cs.stanford.edu>.

*Proceedings of the 42$^{nd}$ International Conference on Machine Learning*, Vancouver, Canada. PMLR 267, 2025. Copyright 2025 by the author(s).

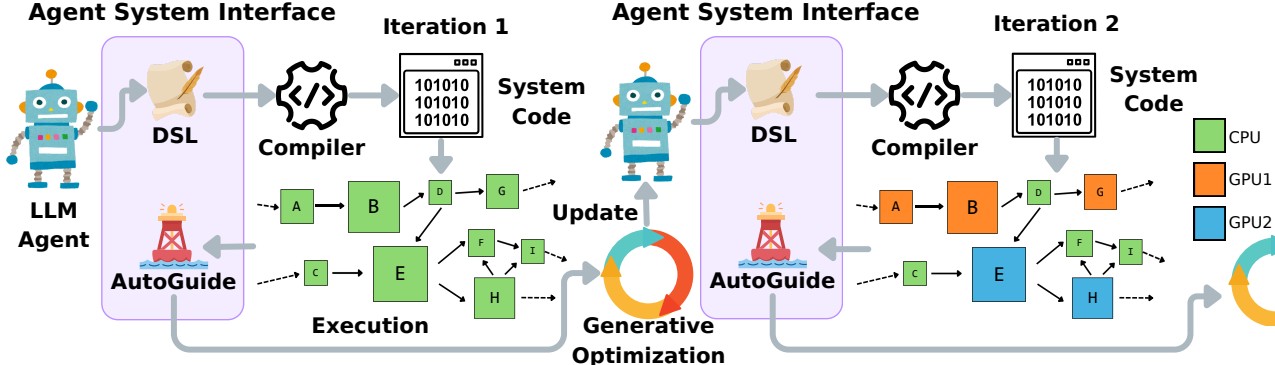

*Figure 1.* **Iterative mapper refinement with agent-based generative optimization.** The system leverages the Agent-System Interface, which consists of the Domain-Specific Language (DSL) and AutoGuide. The DSL abstracts away the low-level system code, defining a search space for mapping strategies, while AutoGuide interprets execution results into actionable guidance. As iterations progress, the mapper evolves to improve performance.

poses the agent to intricate system APIs, coupled with the problem that raw feedback messages from the system are often uninformative to the agent. The second challenge involves *optimizing mapper performance*. Specifically, it consists of (1) defining an appropriate search space and (2) devising efficient methods to find optimal mappers, thereby maximizing parallel program performance.

To address the first challenge, we propose an **Agent-System Interface** (ASI), as shown in Figure 1, an abstraction layer between the agent and the system that simplifies code generation and provides more meaningful feedback to the agent. At the core of ASI is a *Domain-Specific Language (DSL)*, a high-level interface that encapsulates all performance-critical decisions required to generate a mapper. The DSL abstracts away the complexity of low-level system code with a compiler. Additionally, the DSL defines a structured search space, enabling systematic exploration of mapping strategies. We also design and implement the *AutoGuide* mechanism to interpret raw execution output into informative and actionable guidance. This mechanism allows the agent to iteratively optimize the mapper by leveraging enriched feedback to update its strategy.

For the second challenge, we adopt the **generative optimization** approach, a recent advance in optimization techniques. Unlike traditional methods such as reinforcement learning (Ansel et al., 2014), which rely solely on scalar rewards, generative optimization can utilize richer forms of feedback, such as error explanations and actionable suggestions expressed in natural language. This agentic optimization workflow has previously proven to be effective across various domains (Nie et al., 2024; Cheng et al., 2024; Yang et al., 2023; Khattab et al., 2023; Yuksekgonul et al., 2024). Our work is the first to apply such technique to the domain of system optimization.

Our experiments demonstrate that mappers optimized by

LLM-powered agents not only match but often surpass expert-written mappers, achieving up to $1.34\times$ speedup across nine benchmarks. Since expert-written mappers set the highest standard, surpassing them is a notable accomplishment. At the same time, our method significantly reduces mapper tuning time from days to minutes, making high-performance mapping more accessible to domain scientists. To further highlight the advantage of generative optimization, we compare it against OpenTuner, a reinforcement learning-based autotuning framework. Our generative optimizer finds mappers $11\times$ faster than OpenTuner when both run for 10 iterations and still maintains a $3.8\times$ advantage even when OpenTuner runs for 1000 iterations. Furthermore, ablation studies underscore the necessity of the agent-system interface design in achieving these performance gains. Our contributions are as follows:

1. **Design of an Agent-System Interface:** We introduce an abstraction layer that simplifies mapper code generation and provides guidance to the agent. The Domain-Specific Language (DSL) defines a search space, allowing the agent to explore mapping strategies without dealing with low-level system code. AutoGuide interprets raw execution output into targeted feedback, enabling the agent to refine mapper code more effectively.

2. **Generative Optimization for Systems:** We introduce generative optimization to improve system performance, leveraging richer feedback such as error messages and actionable suggestions in natural language. Unlike reinforcement learning methods like OpenTuner, which rely solely on scalar feedback, our method identifies better mappers in far fewer iterations. With only 10 iterations, it outperforms OpenTuner by $3.8\times$ even after 1000 iterations.

3. **Empirical Evaluation of Performance:** Our agent-based solution achieves up to $1.34\times$ speedup across

nine benchmarks, surpassing expert-written mappers while reducing tuning time from days to minutes. We highlight the critical role of the agent-system interface through ablation studies, demonstrating its impact on achieving the performance gains.

## 2. Related Work

**Mapping in Parallel Programming** Many parallel programming systems allow users to make their own mapping decisions, such as Legion (Bauer et al., 2012), StarPU (Augonnet et al., 2009; 2010), Chapel (Chamberlain et al., 2007), HPX (Kaiser et al., 2014; Heller et al., 2017), Sequoia (Fatahalian et al., 2006), Ray (Moritz et al., 2018), TaskFlow (Huang et al., 2021), and Pathways (Barham et al., 2022). Several techniques have been proposed to automate mapping, including machine learning models (O'Boyle et al., 2013; Wang & O'Boyle, 2009), static analysis (Poesia et al., 2017; Ren et al., 2008), reinforcement learning (Ansel et al., 2014; Mirhoseini et al., 2017) and auto-tuning (SFX Teixeira et al., 2023). We use an agent-based approach with LLMs and explore a larger search space for mappers than traditional methods.

**Agentic Frameworks** Agents powered by Large Language Models (LLMs) play a critical role in decision-making, planning, tool integration, and solving complex problems in dynamic environments (Guo et al., 2024). Many agentic frameworks have been developed (Yao et al., 2022; Wu et al., 2023; Li et al., 2023; Hong et al., 2023), with uses spanning domains such as software engineering (Gur et al., 2023; Yang et al., 2024b; Jin et al., 2024), robotics (Kannan et al., 2024), healthcare (Li et al., 2024), education (Ramirez & Esparrell, 2024), and knowledge engineering (Shao et al., 2024). Our work is the first to apply an agentic workflow to iteratively optimize mapper code, improving the performance of parallel programs.

**AI for Systems** The application of AI to optimize system design has gained significant traction in recent years. Techniques such as deep learning (Zheng et al., 2020a;b; 2022b) and gradient-boosted trees (Feng et al., 2023) have been used to predict program execution times for performance optimization. Reinforcement learning methods have addressed challenges in chip floorplanning (Mirhoseini et al., 2021), autotuning (Ansel et al., 2014), auto-vectorization (Haj-Ali et al., 2020a), and compiler phase ordering (Haj-Ali et al., 2020b). While previous efforts have predominantly relied on traditional approaches for cost prediction and optimization, our work uses the recent advances in generative optimization to tackle complex system challenges.

**Generative Optimization** Recent work has explored the use of LLMs for optimization problems traditionally tackled with numerical methods, including mixed-integer programming (AhmadiTeshnizi et al., 2024a;b) and numerical optimization (Nie et al., 2024). A key advantage of generative optimization is its ability to iteratively refine solutions using diverse forms of feedback. For example, Cheng et al. (2024) applies generative optimization to robotic manipulation and game playing, while Yuksekgonul et al. (2024) optimizes prompts and molecular designs. While reinforcement learning has been applied to system optimization, the potential of LLM-driven optimization in systems remains unexplored. Our work explores whether generative optimization with richer feedback outperforms traditional methods using scalar rewards in system optimization.

## 3. Problem Definition

**Motivation and Challenges** The concrete problem we address is the automated generation of high-performance mappers for the Legion parallel programming framework (Bauer et al., 2012). Mappers dictate task scheduling and data placement. A well-designed mapper can achieve orders-of-magnitude speedup over naive strategies.

However, automating mapper generation is challenging due to two key factors. First, **the complexity of low-level system code**. Implementing a mapper requires writing hundreds of lines of intricate C++ code, demanding expertise in system internals. Second, **the vast search space of mapping strategies**. The search space grows exponentially with the number of tasks and arguments.

**Search Space and Performance Impact** As illustrated in Figure A1, the search space of mappers involves multiple decisions, each influencing performance. The first key aspect is **processor selection**, which determines whether a task runs on GPUs, CPUs, or the OpenMP runtime. This choice depends on factors such as task size, GPU memory capacity, and kernel launch overhead. For instance, small tasks may prefer CPUs due to the overhead of launching GPU kernels, while tasks with large memory footprints may run on CPUs when GPU memory is insufficient.

Another crucial dimension is **memory placement**, which dictates where data is stored. A mapper must decide whether to place data in the GPU's FrameBuffer for fast access, ZeroCopy memory for CPU-GPU sharing, or CPU system memory for more available storage. Each option presents trade-offs between access speed, memory usage, and data transfer overhead.

Additionally, **memory layout** further expands the search space, with decisions on Struct of Arrays (SOA) vs. Array of Structures (AOS), data ordering (Fortran-order vs. C-order), and alignment constraints (e.g., 128-byte alignment) significantly affecting cache efficiency and perfor-

```
1  # Map task0 to GPU.
2  Task task0 GPU;
3
4  # Place certain data onto GPU ZeroCopy.
5  Region * ghost_region GPU ZCMEM
6
7  # Specify layout in memory
8  # (aligned to 64 bytes)
9  Layout * * * C_order SOA Align==64
10
11 # Define a cyclic mapping strategy
12 def cyclic(Task task):
13     ip = task.ipoint;
14     mgpu = Machine(GPU);
15     node_idx = ip[0] % mgpu.size[0];
16     gpu_idx  = ip[0] % mgpu.size[1];
17     return mgpu[node_idx, gpu_idx];
18
19 IndexTaskMap task4 cyclic
```

```
1  void slice_task(const Task& task,
2                  const SliceTaskInput &input,
3                  SliceTaskOutput &output) {
4    vector<Processor> targets =
5      this->select_targets_for_task(ctx, task);
6    DomainT<2> space = input.domain;
7    Point<2> num_points =
8        space.bounds.hi - space.bounds.lo + ones;
9    Rect<2> blocks(zeroes, num_blocks - ones);
10   ... // 126 lines of C++ code omitted here
11   for (PointInRectIterator<2> it(blocks); it() != NULL; it++)
12   {
13     DomainT<2,coord_t> slice_space;
14     TaskSlice slice;
15     slice.domain = {slice_lo, slice_hi};
16     slice.proc = targets[index++ % targets.size()];
17     output.slices.push_back(slice);
18   }
19 }
```

(a) An example mapper in Domain-Specific Language (DSL)

(b) Code snippet from a C++ mapper

*Figure 2.* **Comparison of a DSL mapper and a C++ mapper.** The DSL's **declarative, high-level** design abstracts away the complexity of low-level C++ code, serving as the core of the **Agent-System Interface**. The highlighted boxes illustrate how the same functionality, which requires extensive C++ system code, can be expressed concisely in just a few lines in DSL.

mance.

Finally, an important idiom in high-performance computing is launching tasks over partitioned data. **Index mapping** determines how data partitions and task executions are distributed across multiple processors. For consistency, we can represent data partitioning as a tensor of data partitions, the machine as a tensor of processors, and tasks operating on the partitioned data as a tensor of tasks. The way data and task indices are mapped to processor indices affects inter-processor communication, a key factor in performance (Unger et al., 2022; Zheng et al., 2022a).

## 4. Our Approach: Agent-System Interface

### 4.1. Domain-Specific Language Design

A key challenge in automating mapper generation with a coding agent is the complexity of low-level system code, which requires intricate C++ implementations. To address this, we design a high-level **Domain-Specific Language (DSL)** as the core of our **Agent-System Interface** (ASI). The DSL provides a *structured search space* for mapping strategies while *abstracting away low-level implementation details*. Unlike C++, which demands imperative specifications of mapping policies, our DSL adopts a **declarative design**, allowing users to specify *what* to achieve rather than *how* to implement it. Most critically, the DSL **separates concerns**, enabling multiple aspects of mapping decisions to be expressed *independently rather than being entangled* in low-level system APIs. This design reduces code complexity and naturally provides a search space for the agent to explore. To implement it, we develop a *compiler* that translates DSL into the low-level C++ APIs.

As illustrated in Figure 2, the complexity of DSL code is

significantly lower than that of C++. Figure 2a provides an example of a DSL mapper, highlighting the key features of our DSL. In contrast, Figure 2b shows a snippet from a C++ mapper, emphasizing the intricacy of low-level implementation details. Across the benchmarks, using the DSL results in an **average lines of code reduction of** $14\times$. This substantial reduction makes DSL a more suitable target for LLM code generation, as it abstracts away the complexities inherent in low-level systems. As we will show in Section 5.2, LLMs generate DSL code more effectively, despite DSL having no examples in LLM training corpora, whereas C++ is widely represented.

Next, we describe the DSL's design, emphasizing its declarative nature and structured search space. Section 3 details the performance impact of each decision.

**The `Task` statement (Line 2)** defines processor selection for each task, choosing between CPU, GPU, or OpenMP. Line 2 specifies that instances of `task0` should run on GPUs. This decision is made per task; note that the search space expands exponentially with the number of tasks.

**The `Region` statement (Line 5)** controls memory placement for data arguments. Line 5 specifies that all tasks using `ghost_region` should place the data in GPU Zero-Copy memory. Other choices include GPU FrameBuffer memory and CPU System Memory. This decision is made per task and per argument, causing the search space to grow exponentially.

**The `Layout` statement (Line 9)** defines memory layouts. Line 9 enforces a `C_order` axis ordering, an `SOA` layout, and a 64-byte memory alignment for all data used by all tasks mapped to all processors. Alternative choices include `F_order`, `AOS`, and various alignment strategies. This is a per-task, per-data, per-processor decision.

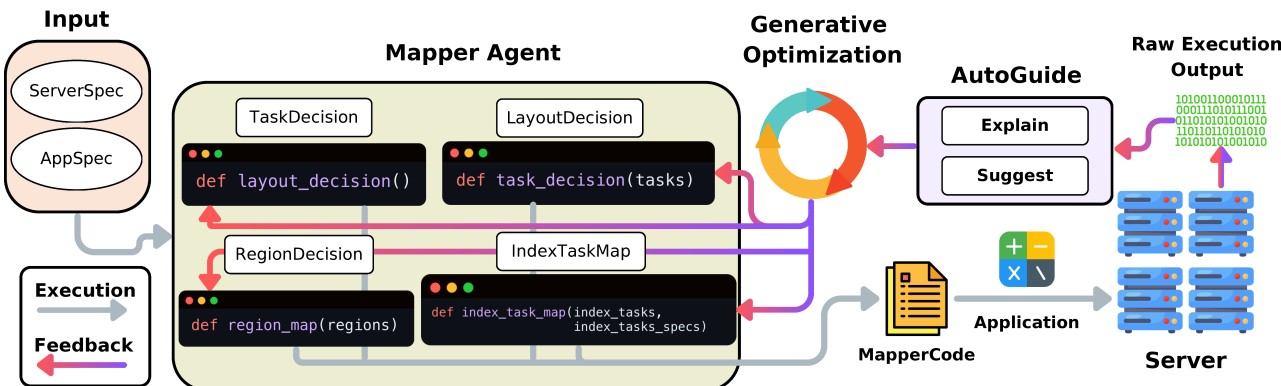

*Figure 3.* **Agent Optimization Process**. The mapper agent takes server specifications and application-specific information as input, generates mapper code, and executes it alongside the application on the server. Raw execution feedback is enriched using the **AutoGuide** mechanism, and the mapper is iteratively refined by an LLM optimizer to improve performance.

| Case | Raw Execution Output | AutoGuide | |
|------|----------------------|-----------|---|
| | | **Explain** | **Suggest** |
| Case 1 | **Execution Error:** Assertion failed: stride does not match expected value. | Memory layout is unexpected. | Adjust the layout constraints or move tasks to different processor types. |
| Case 2 | **Performance Metric:** Execution time is 0.03s. | N/A | Move more tasks to GPU to reduce execution time. |

*Table 1.* **AutoGuide Feedback Mechanism.** The AutoGuide mechanism interprets raw execution output from the runtime system, providing more informative error explanations and suggestions for mapper modifications. It is implemented via keyword matching. Additional examples are shown in Table A3.

**The `IndexTaskMap` statement (Line 19)** controls index mapping using a customized function. Line 12 defines the mapping function that establishes the correspondence between two index spaces: the task index space (represented by `task.ipoint`) defined in the application code (e.g., for loops) and the processor space of the distributed machine (represented by `Machine(GPU)`). The DSL allows users to express arbitrary arithmetic mappings between the two index spaces. This decision applies to each task group launched by parallel for loops.

Our DSL is designed to express a wide range of high-performance mapping strategies, including all of the most important decisions. While there may be cases where certain optimizations are not directly expressible, we have not encountered any. Despite being more constrained than general-purpose C++, the DSL has been proven to be effective: all mappers discovered by our agent that outperform expert-written C++ implementations are expressible within the current DSL.

### 4.2. Generative Optimization via AutoGuide

We formulate mapper generation as an **online optimization problem**. Given a triplet $(\Theta, \omega, \mathcal{T})$, where $\Theta$ is a set of possible mappers, $\omega$ is an *optimization objective*, and $\mathcal{T}$ is a function that takes a mapper $\theta \in \Theta$ as input, $(f, g) = \mathcal{T}(\theta)$ and returns $f$, the *feedback* from executing the mapper (i.e., the measured performance after running the application code with the generated mapper), and $g$, the *process graph* tracing how the mapper was generated. In our setup, mapper performance is deterministic, as we carefully control all sources of randomness in the environment. If the parameter space were numerical, this online optimization problem could be addressed using bandit algorithms (Lattimore & Szepesvári, 2020), reinforcement learning (Sutton & Barto, 2018), or Bayesian optimization (Snoek et al., 2012), but these methods are less efficient when the parameter search space is large and discrete (i.e., text).

In this online optimization problem, we leverage the DSL to structure the parameter space to improve the efficiency of optimization. Here, $\theta$ represents the program code, while $\omega$ and $f$ are expressed as text. We adopt **generative optimization**, leveraging LLMs as optimizers given the objective in text form. This emergent optimization behavior has been recently observed and applied across various domains (Yang et al., 2024a; Cheng et al., 2024; Yuksekgonul et al., 2024; Patel et al., 2024).

**Optimization Process**  We present the optimization process in Figure 3. The agent takes two inputs: server specifications and application metadata. Server specifications detail the hardware configuration, including the number of CPUs and GPUs per node, as well as the total node count. Application metadata provides information on task names and the associated data arguments accessed by each task. These inputs define the structured search space explored by the agent during optimization. The agent, using the given inputs, generates mapper code that is executed alongside the application code on the server. Raw execution feedback from the runtime is augmented with the AutoGuide mechanism and fed back to the LLM, iteratively refining the agent for improved mapper code generation.

**Coding Agent**  Our mapper agent improves mapping decisions by iteratively generating DSL code. A high-level schema of the mapper agent is shown in Figure 3. The mapper agent is implemented as a Python program in the Trace (Cheng et al., 2024) framework, where we decompose the task of generating a monolithic mapper into *independent code segments*. This decomposition allows the agent to decide what code to generate for each segment separately. This approach is effective because our *DSL design eliminates unnecessary dependencies* between mapping decisions. Our modularization strategy aligns with least-to-most prompting (Zhou et al., 2022).

**AutoGuide**  The AutoGuide feedback mechanism is designed based on three key motivations: (1) generative optimization benefits from natural language feedback rather than relying solely on scalar values, (2) raw execution output from the runtime system is often too uninformative to effectively guide the agent's decisions, and (3) domain heuristics known to systems researchers can be naturally expressed in language (e.g., most tasks run faster on GPUs than CPUs). To address these needs, AutoGuide helps the agent by **explaining** opaque error messages and **suggesting** mapper modifications. As shown in Table 1, it interprets uninformative execution output into actionable insights, with additional examples in Appendix A.5. The implementation relies on keyword matching over the raw execution output. An ablation study in Section 5.3 demonstrates its effectiveness in our experiments.

## 5. Evaluation

Experiments are conducted on one node with two Intel 10-core E5-2640 v4 CPUs, 256G main memory, and four NVIDIA Tesla P100 GPUs. We use gpt-4o-2024-08-06.

### 5.1. Speedup of Application Performance

**Benchmarks**  Our evaluation utilizes a suite of 9 benchmarks, including 3 scientific computing workloads and 6 well-known matrix multiplication algorithms. *Circuit* is a simulation benchmark that models electrical circuit behavior by simulating currents and voltages across interconnected nodes and wires (Bauer et al., 2012). *Stencil* simulates a 2D grid where each point's value is updated based on a stencil pattern determined by its neighbors (Van der Wijngaart & Mattson, 2014). *Pennant* models unstructured mesh Lagrangian staggered-grid hydrodynamics, commonly used for simulating compressible flow (Ferenbaugh, 2015). The remaining six benchmarks – *Cannon's*, *SUMMA*, *PUMMA*, *Johnson's*, *Solomonik's*, and *COSMA* – are well-known parallel matrix multiplication algorithms (Cannon, 1969; Van De Geijn & Watts, 1997; Choi et al., 1994; Agarwal et al., 1995; Solomonik & Demmel, 2011; Kwasniewski et al., 2019). Parallel matrix multiplication remains an active research topic due to its central role in high-performance computing and scientific simulations (Yadav et al., 2022). Furthermore, improving matrix multiplication performance has a broad impact, as it accelerates numerous downstream machine learning workloads (Jangda et al., 2022; Zheng et al., 2025). We discuss these matrix multiplication algorithms in more detail in Appendix A.3. This benchmark suite provides both depth with its representative matrix multiplication algorithms and variety with its range of scientific computing workloads.

In this experiment, we evaluate the performance of the mappers with the following baselines.

**Expert-Written Mappers.**  These mappers are manually developed by domain scientists who spend years mastering computational science. Writing mappers in parallel programming frameworks is another challenge, and tuning them for specific applications can take days.

**Randomly Generated Mappers.**  These mappers were randomly generated with 10 different random seeds, sampling from the entire search space of each application. We report the average performance.

**Agent-Optimized Mappers.**  Using Trace (Cheng et al., 2024), we evaluated the **Trace** and **OPRO** (Yang et al., 2023) search algorithms, running 10 iterations per application. To account for stochastic output, we repeated the process 5 times and report the average. The best mapper from Trace across runs is also reported.

**OpenTuner Mappers.**  OpenTuner (Ansel et al., 2014) is a program autotuning framework that uses reinforcement learning to optimize performance based on scalar feedback. We provided execution time as feedback, with a high penalty for failures.

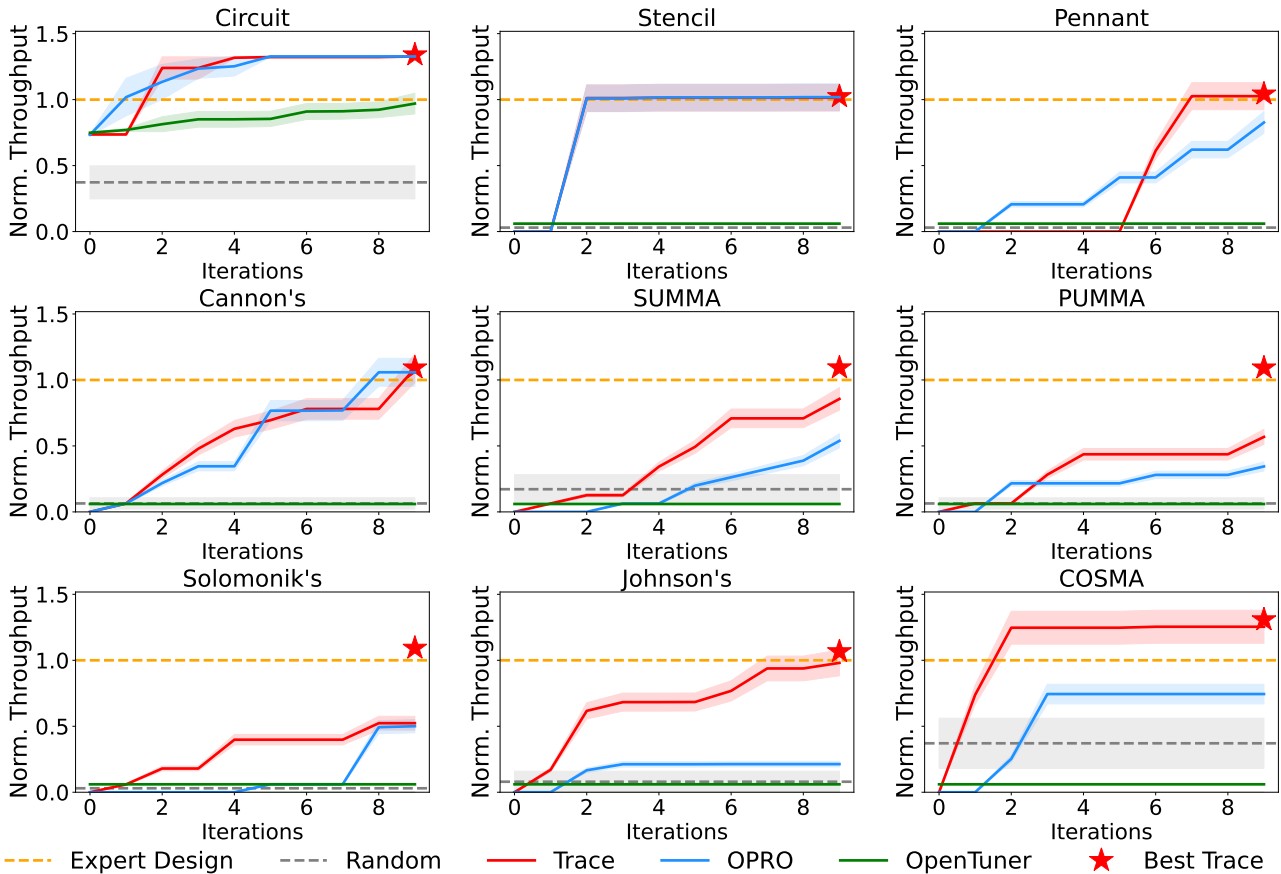

Figure 4. **Performance Comparison.** Normalized throughput for 9 benchmarks, comparing expert mappers, random mappers, the average optimization trajectories of Trace, OPRO, and OpenTuner in 10 iterations across 5 runs, and the best mappers found by Trace.

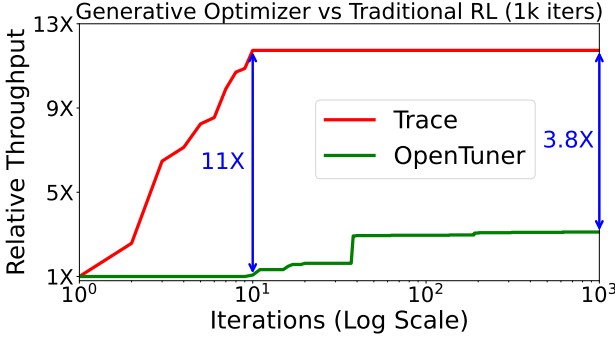

Figure 5. Comparison of Trace (generative optimizer) and Open-Tuner (traditional RL) over 1K iterations (averaged across all 9 benchmarks)

**Results** In Figure 4, we use normalized throughput as our performance metric, where higher values indicate better performance. The throughput is normalized relative to the expert-written mappers, providing a clear baseline for comparison. Our focus is on measuring end-to-end perfor-

mance, which includes both the correctness and efficiency of the generated mappers. If the generated code has any syntax or runtime issues, its throughput is recorded as 0. We report the best mappers found by Trace, and the average optimization trajectories of Trace, OPRO and Open-Tuner over 10 iterations across 5 runs.

**All the best mappers found by Trace can match or surpass the expert-written mappers**, underscoring the effectiveness of agent-based generative optimizer. In our context, reporting the best-performing mapper is appropriate. Mapper optimization is an offline process, and in practice, it is standard to run the optimizer multiple times and deploy the best result. Once identified, the mapper can be reused across repeated executions on the same application, input, and hardware, incurring no further search cost.

Random mappers consistently exhibit low performance across all applications, emphasizing the critical role of mapping decisions. For each application, we generate 10 random mappers by sampling from the full DSL-defined search space, totaling 90 mappers across 9 applications. Among them, 74 (82.2%) raise runtime errors due to invalid

| Code Generation Target | Mapping Strategy | | | | | | | | | | Success Rate |
|---|---|---|---|---|---|---|---|---|---|---|---|
| | 1 | 2 | 3 | 4 | 5 | 6 | 7 | 8 | 9 | 10 | |
| C++ (single trial) | ✗ | – | – | ✗ | – | – | ✗ | ✗ | – | – | 0% |
| DSL (single trial) | ✓ | ✓ | ✓ | ✓ | ✓ | – | ✓ | ✓ | ✓ | – | 80% |
| C++ (iterative refine) | ✗ | – | – | ✗ | ✗ | ✗ | ✗ | ✗ | ✗ | ✗ | 0% |
| DSL (iterative refine) | ✓ | ✓ | ✓ | ✓ | ✓ | ✓ | ✓ | ✓ | ✓ | ✓ | 100% |

*Table 2.* **Code Generation Success Rates.** Success rates for generating code across 10 mapping strategies described in natural language. The test evaluates whether the generated code compiles and passes execution tests. Generating DSL code significantly outperforms generating C++ for both settings. Symbols indicate results: – fails to compile, ✗ compiles but fails the test, and ✓ passes the test.

mapping decisions. The runtime system enforces correctness by rejecting such mappers during execution, resulting in a throughput of 0.

When comparing optimization trajectories, Trace performs similarly to OPRO, and significantly outperforms Open-Tuner. To further compare the agent-based optimizer with traditional reinforcement learning, we extended Open-Tuner's optimization iterations from 10 to 1000, as shown in Figure 5, where the x-axis is the log-scale of iterations and the y-axis represents relative throughput (averaged across all 9 benchmarks). Notably, Trace achieves a $3.8\times$ speedup over OpenTuner even when OpenTuner is run for 1000 iterations. When both are limited to 10 iterations, Trace outperforms OpenTuner by $11\times$, demonstrating its ability to quickly identify high-performance mappings. **This highlights the superiority of Trace (generative optimizer) over OpenTuner (traditional reinforcement learning).** Moreover, Trace completes the entire optimization process in just 10 minutes per application, **reducing mapper development time from days to minutes**.

To offer a more comprehensive view of performance variations, we present additional statistics, including the mean, standard deviation, worst, median, and best normalized throughput for both our method and OpenTuner. These statistics are derived from five runs for each benchmark, as detailed in Appendix A.4.

**Case Analysis** The largest performance gain achieved by Trace over the expert mapper is observed in Circuit, with a speedup of $1.34\times$. This improvement is primarily due to *memory placement*: the best mapper allocates two data collections to GPU FrameBuffer memory, while the expert mapper places them in GPU ZeroCopy memory. Despite a slight increase in inter-GPU communication costs, Trace reduces task execution time due to faster memory access, resulting in higher overall performance. For matrix-multiplication algorithms, the greatest speedup is seen in COSMA, with Trace achieving a $1.31\times$ speedup over the expert mapper. This is attributed to Trace's more efficient index mapping functions, which *reduce inter-GPU communication* by better distributing partitioned submatrices

across GPUs. For additional context, examples of Trace mappers are presented in Appendix A.8.

### 5.2. Ablation Study of DSL for Code Generation

In Section 5.1, we demonstrate the overall effectiveness of our approach. Here, we conduct an ablation study on the DSL, the core of the Agent-System Interface. Since successful generation is the foundation of optimization, this subsection focuses on **how well the DSL helps LLMs *generate* syntactically and semantically correct mappers compared to C++**, rather than directly *optimizing* performance.

**Experiment Setup** We designed 10 mapping strategies, described in natural language, to evaluate whether LLMs can generate correct code in both the DSL and the original low-level C++. The strategies are detailed in Appendix A.7. To ensure a fair comparison, identical prompt materials (documentation, examples, and starting code) were provided for both the DSL and C++. Success rates are measured based on whether the generated code passes predefined test cases, with results reported for single trials and iterative refinement, where the LLM is allowed up to 10 iterations to improve the code using compiler feedback. The evaluation is conducted with the DSPy (Khattab et al., 2023) framework.

**Results** Table 2 shows that **DSL achieves significantly higher generation success rates than C++** in both the single-trial and iterative refinement settings. This demonstrates the effectiveness of DSL's design in abstracting system complexity and providing a high-level interface that enables LLMs to tackle complex system challenges in code generation. Incorporating iterative refinement with compiler feedback further improves success rates, resolving four compilation errors in C++ and two in the DSL. However, the gap between DSL mappers and C++ mappers remains substantial. Notably, these results are striking given that the DSL is a low-resource language with no pre-training or fine-tuning data, while C++ code is widely present in LLM training corpora.

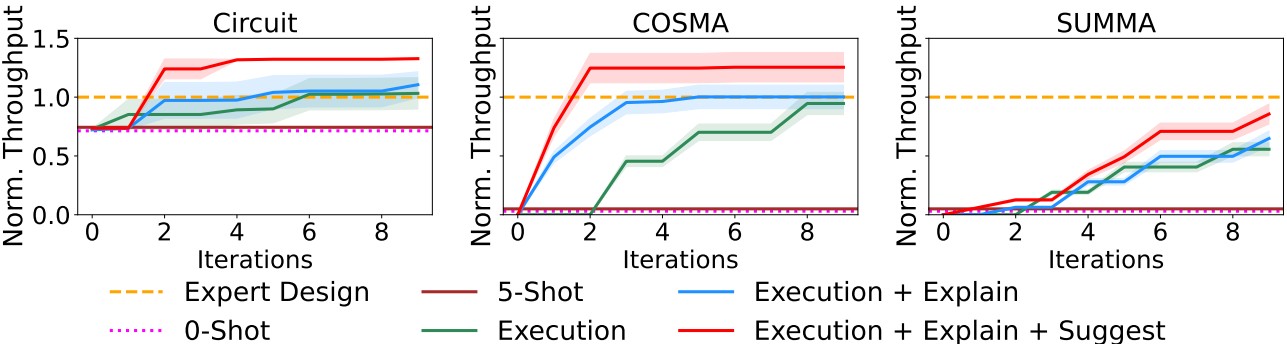

*Figure 6.* **Comparison of different feedback designs. 0-Shot** and **5-Shot** are baselines. **Execution** provides only the raw execution output as feedback. **Explain** provides additional explanations of execution errors. **Suggest** offers mapper modification suggestions. All feedback is automatically generated.

**Analysis**  LLMs perform better with the DSL for two reasons. First, the **semantic gap** between natural language and code is smaller with the DSL than with C++. For example, writing a mapper to "align all data to 64 bytes in memory and use Fortran ordering" requires one line `Layout * * * Align==64 F_order` in the DSL because of its *declarative design*. In contrast, the C++ mapping API requires a sequence of operations to enforce alignment and ordering, which widens the semantic gap. Second, the DSL reduces the **amount of code**. As discussed before, LLMs achieve an average reduction of $14\times$ in lines of code, simplifying code generation. These results underscore the importance of a high-level agent-system interface.

### 5.3. Ablation Study of the AutoGuide Feedback

The AutoGuide mechanism provides enriched feedback to the agentic optimizer. We compare with alternative feedback designs.

**Experiment Setup**  We compare the following baselines. **0-shot** and **5-shot** have no feedback, allowing the LLM to generate once with either 0 or 5 examples provided. **Execution** only provides raw execution feedback, **Explain** offers additional explanations for execution errors, and **Suggest** offers mapper modification suggestions. The Trace trajectory shown in Figure 4 uses the full AutoGuide mode with all **Execution+Explain+Suggest**. As an ablation study, we evaluate 3 benchmarks.

**Results and Analysis**  Figure 6 demonstrates that the **full feedback mechanism consistently outperforms** all reduced feedback variants. The 0-shot and 5-shot results perform the worst, underscoring the importance of feedback-based iterative refinement. This highlights the value of an agentic workflow, showing that performance improvements are not solely driven by prompting the LLM but are a direct result of the iterative refinement in the workflow design.

## 6. Conclusion

In this paper, we introduced a system that leverages LLMs to automate mapper generation and optimization. The Agent-System Interface (ASI) simplifies code generation with a Domain-Specific Language (DSL), which abstracts away the low-level complexity of system code, and enriches execution feedback through AutoGuide, which interprets raw execution output into actionable guidance. We adopted generative optimization, allowing LLMs to refine mappers using rich textual feedback beyond scalar metrics. Unlike RL-based methods like OpenTuner, which rely on numerical rewards, our approach incorporates error explanations and targeted suggestions, accelerating search efficiency. Experiments show that agent-generated mappers outperform expert-written ones, achieving up to $1.34\times$ speedup across nine benchmarks. Our method, running only 10 iterations, maintains a $3.8\times$ advantage over Open-Tuner even after 1000 iterations. By reducing mapper development time from days to minutes, our approach benefits computational scientists and demonstrates the effectiveness of generative optimization in system design.

## Acknowledgements

This work was supported in part by a Google Research Award. We thank Yuhui Zhang, Genghan Zhang, Qizheng Zhang, Mohammad R. Fadiheh, Ed Chen, Mert Yuksek-gonul, Chenyang Yang, Jiaxin Ge, Simon Guo, and Anne Ouyang for the feedback. We thank Ching-An Cheng and Ken Liu for their discussions and feedback.

## Impact Statement

This paper presents work whose goal is to advance the field of Machine Learning. There are many potential societal consequences of our work, none of which we feel must be specifically highlighted here.

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

# A. Appendix

## A.1. Illustration for Mapping

We show an illustration for mapping in Figure A1.

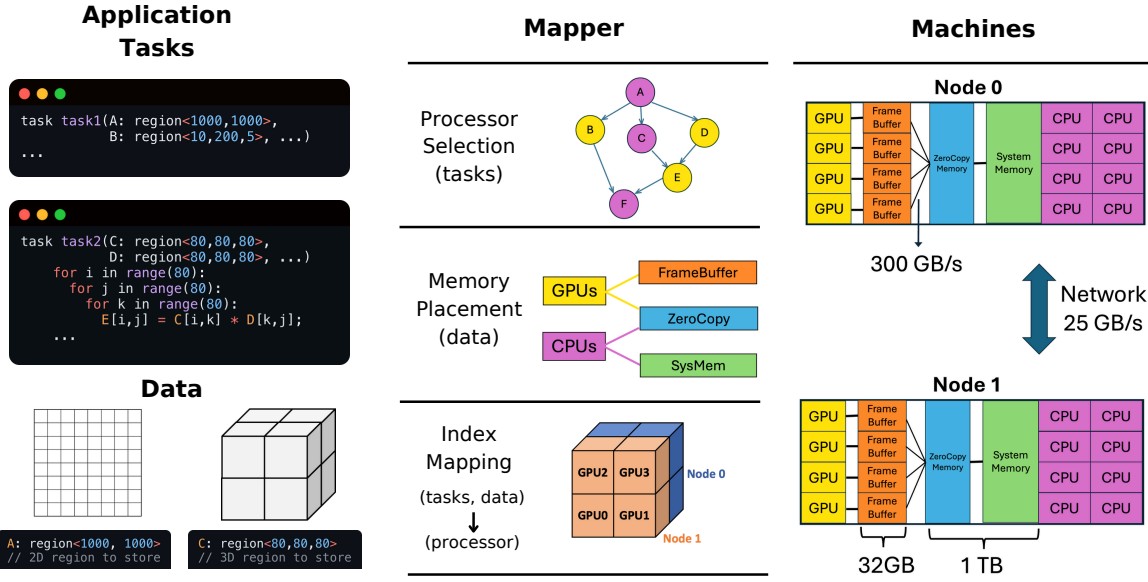

*Figure A1.* Mappers decide the placement of each task in the task graph to processors, the placement of data to memory, and how the iteration space of data is partitioned and mapped to different processors.

## A.2. DSL Grammar

**Terminals:** `TaskName`, `RegionName`, `var`, `int`

**Grammar Rules:**

```
Program     →   Statement+
Statement   →   TaskMap | DataMap | DataLayout | FuncDef | IndexTaskMap TaskName var
TaskMap     →   Task TaskName Proc+
DataMap     →   Region TaskName RegionName Proc Memory+
Proc        →   CPU | GPU | OMP
Memory      →   SYSMEM | FBMEM | ZCMEM
DataLayout  →   Layout TaskName RegionName Proc Constraint+
Constraint  →   SOA | AOS | C_order | F_order | Align == int
FuncDef     →   def var(var+):  FuncStmt+
FuncStmt    →   var = Expr | return Expr
Expr        →   var | var(Expr+) | Machine(Proc) | Expr.Expr | Expr Op Expr | (Expr) |
                Expr[Expr] | *Expr | Expr ?  Expr :  Expr
```

### A.3. Parallel Matrix Multiplication Algorithms

**2D Algorithms**  Cannon's (Cannon, 1969) introduced a systolic communication pattern with tiled data partitioning for distributed matrix multiplication. PUMMA (Choi et al., 1994) and SUMMA (Van De Geijn & Watts, 1997) extended this approach by supporting non-square matrices and improving communication efficiency through pipelining. They are called 2D algorithms because they partition the matrices into 2D tiles and then map them onto the processor space.

**Non-2D Algorithms**  Johnson's (Agarwal et al., 1995) introduced a 3D algorithm that partitions the input matrices into 3D tiles and uses additional memory per processor to reduce communication compared to 2D algorithms. Solomonik's (Solomonik & Demmel, 2011) balances between 2D and 3D approaches by using extra memory to further minimize communication. COSMA (Kwasniewski et al., 2019) takes a different approach by optimizing the processor grid and parallelization strategy based on the input size and the machine size.

### A.4. Additional Performance Statistics

In our setting, reporting the best result across multiple runs is appropriate, as the best mapper is the one that is desired by the user. Mapper search is an offline optimization process, and it is feasible to run the optimizer multiple times to select the highest-performing mapper. Once identified, this mapper can be reused without incurring additional search cost, as the deployment scenario (application, input, and hardware) remains fixed.

That said, additional statistics on performance varations can provide a more complete picture. Here we include the mean, standard deviation, worst, median, and best normalized throughput across five runs for each benchmark of our approach Trace and OpenTuner.

| Benchmark | Mean | Std Dev | Worst | Median | Best |
|---|---|---|---|---|---|
| Circuit | 1.33× | 0.01 | 1.31× | 1.33× | 1.34× |
| Stencil | 1.01× | 0.01 | 1.00× | 1.01× | 1.02× |
| Pennant | 1.03× | 0.02 | 1.00× | 1.03× | 1.04× |
| Cannon | 1.09× | 0.00 | 1.08× | 1.09× | 1.09× |
| SUMMA | 0.86× | 0.48 | 0.00× | 1.07× | 1.09× |
| PUMMA | 0.57× | 0.55 | 0.00× | 0.66× | 1.09× |
| Johnson | 0.98× | 0.17 | 0.68× | 1.06× | 1.07× |
| Solomonik | 0.52× | 0.41 | 0.00× | 0.61× | 1.09× |
| COSMA | 1.25× | 0.03 | 1.23× | 1.23× | 1.31× |

*Table A1.* Normalized throughput of our framework Trace.

| Benchmark | Mean | Std Dev | Worst | Median | Best |
|---|---|---|---|---|---|
| Circuit | 0.97× | 0.16 | 0.81× | 0.99× | 1.20× |
| Stencil | 0.00× | 0.00 | 0.00× | 0.00× | 0.00× |
| Pennant | 0.00× | 0.00 | 0.00× | 0.00× | 0.00× |
| Cannon | 0.00× | 0.00 | 0.00× | 0.00× | 0.00× |
| SUMMA | 0.00× | 0.00 | 0.00× | 0.00× | 0.00× |
| PUMMA | 0.00× | 0.00 | 0.00× | 0.00× | 0.00× |
| Johnson | 0.00× | 0.00 | 0.00× | 0.00× | 0.00× |
| Solomonik | 0.00× | 0.00 | 0.00× | 0.00× | 0.00× |
| COSMA | 0.00× | 0.00 | 0.00× | 0.00× | 0.00× |

*Table A2.* Normalized throughput of OpenTuner.

Our method achieves relatively stable performance across most benchmarks. The higher variance and occasional 0.00× worst-case throughput observed in SUMMA, PUMMA, and Solomonik are due to invalid mapper configurations in the search space (e.g., violating cuBLAS layout constraints). The runtime enforces correctness by rejecting such configurations during execution. While the generative optimizer typically learns to avoid these cases through the AutoGuide mechanism, occasional failures within the 10-iteration budget are still possible. In practice, such failures can be mitigated by repeating the optimization and selecting the best-performing mapper. In contrast, OpenTuner, despite running the same number of iterations, fails to generate valid mappers for 8 out of 9 benchmarks. This highlights the difficulty of exploring the search space using traditional reinforcement learning methods.

## A.5. Examples of Feedback Configurations

We give examples for the raw execution output and enriched feedback in Table A3. The enhanced feedback includes explanations of errors and suggestions for mapper modifications.

| Case | Raw Execution Output | AutoGuide | |
| --- | --- | --- | --- |
| | | **Explain** | **Suggest** |
| case1 | **Compile Error:** Syntax error, unexpected :, expecting { | N/A | There should be no colon : in function definition. |
| case2 | **Compile Error:** IndexTaskMap's function undefined | N/A | Define the IndexTaskMap function first before using it. |
| case3 | **Compile Error:** mgpu not found | N/A | Include `mgpu = Machine(GPU);` in the generated code. |
| case4 | **Execution Error:** Assertion failed: stride does not match expected value. | Memory layout is unexpected. | Adjust the layout constraints or move tasks to different processor types. |
| case5 | **Execution Error:** DGEMM parameter number 8 had an illegal value | Memory layout is unexpected. | Adjust the layout constraint. |
| case6 | **Execution Error:** Slice processor index out of bound | IndexTaskMap statements cause error. | Ensure that the first index of mgpu ends with `% mgpu.size[0]`, and the second element ends with `% mgpu.size[1]`. |
| case7 | **Execution Error:** Assertion 'event.exists()' failed | InstanceLimit statements cause error. | Avoid generating InstanceLimit statements. |
| case8 | **Performance Metric:** Execution time is 0.03s. | N/A | Move more tasks to GPU to reduce execution time. |
| case9 | **Performance Metric:** Achieved throughput = 4877 GFLOPS | N/A | Try using different `IndexTaskMap` or `SingleTaskMap` statements to maximize throughput. |

*Table A3.* Raw execution output and AutoGuide (error explanations and adjustment suggestions) for different cases.

## A.6. Trace Agent Code

Trace (Cheng et al., 2024) uses Python decorators like `@bundle` to annotate Python programs. It allows us to design an LLM code generation agent as if we were writing a Python program ourselves. We first set up an end-to-end runnable Python program that can generate a valid mapper program by randomly making decisions over the search space. We show the high-level structure of our Trace Mapper in Figure A3. Figure A2 shows how we incorporate the feedback from the execution to update the agent. At each optimization step, Trace will execute `DSLMapperGenerator` and collect the corresponding execution flow to build up a graph. Then it will make a call to an LLM to perform an update to any function that is decorated with `@bundle(trainable=True)`. The `DSLMapperGenerator` is structured in the same way as providing a search space specified by the DSL, where an LLM optimizer can make decisions along the pre-designed axes. We note that this type of design is only enabled by recent developments like Trace and is much more challenging to do using older LLM-based frameworks.

```python
1  policy = MapperAgent()
2  params = policy.parameters()
3  optimizer = trace.Optimizer(params)
4
5  app  = GetApplicationInfo()
6  test = GetMapperEvaluator(app)
7
8  for i in range(iterations):
9    # Forward pass
10   try:
11     mapper = policy(app)
12     # feedback (str) contains performance
13     feedback = test(mapper)
14   except TraceExecutionError as e:
15     feedback = str(e)
16     target = e.exception_node
17
18   # Backward pass and update
19   optimizer.zero_feedback()
20   optimizer.backward(target, feedback)
21   optimizer.step()
```

*Figure A2.* We show how we use Trace to incorporate the feedback from the execution to update the agent, with a Pytorch-like syntax.

```python
import opto.trace as trace

class MapperAgent(trace.Module):
    @trace.bundle(trainable=True)
    def task_decision(self, tasks):
        ...

    @trace.bundle(trainable=True)
    def region_decision(self, regions):
        ...

    @trace.bundle(trainable=True)
    def layout_decision(self):
        ...

    @trace.bundle(trainable=True)
    def instance_limit_decision(self, tasks):
        ...

    @trace.bundle(trainable=True)
    def index_task_map_decision(self, index_tasks):

    @trace.bundle(trainable=True)
    def single_task_map_decision(self, single_tasks):
        ...

    def generate_mapper(self):
        """
        Generate the final mapper code by combining all code statements.
        """
        task_statements = self.task_decision(self.tasks)
        region_statements = self.region_decision(self.regions)
        layout_statements = self.layout_decision()
        instance_limit_statements = self.instance_limit_decision(self.tasks)
        index_task_map_statements = self.index_task_map_decision(self.index_tasks,
    self.index_task_specification)
        single_task_statements = self.single_task_map_decision(self.single_tasks)

        code_statements = (
            task_statements +
            region_statements +
            layout_statements +
            instance_limit_statements +
            index_task_map_statements +
            single_task_statements
        )
        # Combine all code statements and function definitions into a single string
        code_list = code_statements
        mapper_code = str_join(node('\n'), *code_list)
        return mapper_code
```

*Figure A3.* High-level structure of the Trace-based agent template, where functions annotated with `@bundle(trainable=True)` define the search space that the LLM optimizer updates during mapper generation. **Note**: This agent serves as a shared starting point for **ALL** tasks. For each task, we produce a mapper from this starting agent and then ask LLMs to "optimize" this agent (by changing functions that are `trainable`) to produce mappers that are optimal for the particular task.

### A.7. Mapping Strategies

**Strategy 1:** Map the tasks of `calculate_new_currents`, `distribute_charge`, `update_voltages` onto GPUs in this way: linearize the 2D GPU processor space into 1D, then perform 1D block mapping from launch domain to the linearized 1D processor space.

```
1 Task * GPU,CPU; # for any task, run on GPU if supported
2 Region * *GPU FBMEM; # for any task, any region, if mapped onto GPU, use FBMEM as default
3 Region * * CPU SYSMEM; # if mapped onto CPU, use SYSMEM as default
4
5 Layout * * * SOA C_order;
6
7 mcpu = Machine(CPU);
8 mgpu = Machine(GPU);
9
10 ========== Above is fixed ==========
11 def linearblock(Task task) {
12     return mgpu[task.ipoint[0] / mgpu.size[1], task.ipoint[0] % mgpu.size[1]];
13 }
14
15 IndexTaskMap calculate_new_currents,distribute_charge,update_voltages linearblock;
```

**Strategy 2:** Place ghost/shared regions (rp_shared and rp_ghost) onto GPU zero-copy memory

```
1 Task * GPU,CPU; # for any task, run on GPU if supported
2
3 Region * * GPU FBMEM; # for any task, any region, if mapped onto GPU, use FBMEM as default
4 Region * * CPU SYSMEM; # if mapped onto CPU, use SYSMEM as default
5
6 Layout * * * SOA C_order;
7
8 mcpu = Machine(CPU);
9 mgpu = Machine(GPU);
10
11 ========== Above is fixed ==========
12
13 Region * rp_shared GPU ZCMEM;
14 Region * rp_ghost GPU ZCMEM;
```

**Strategy 3**: Use Array Of Struct (AOS) data layout for all data instead of the default SOA

```
1 Task * GPU,CPU; # for any task, run on GPU if supported
2
3 Region * * GPU FBMEM; # for any task, any region, if mapped onto GPU, use FBMEM as default
4 Region * * CPU SYSMEM; # if mapped onto CPU, use SYSMEM as default
5
6 mcpu = Machine(CPU);
7 mgpu = Machine(GPU);
8
9 ========== Above is fixed ==========
10
11 Layout * * * AOS;
```

**Strategy 4**: Use Fortran ordering of data layout for all data instead of the default C order

```
1 Task * GPU,CPU; # for any task, run on GPU if supported
2
3 Region * * GPU FBMEM; # for any task, any region, if mapped onto GPU, use FBMEM as default
4 Region * * CPU SYSMEM; # if mapped onto CPU, use SYSMEM as default
5
6 mcpu = Machine(CPU);
7 mgpu = Machine(GPU);
8
9 ========== Above is fixed ==========
10
11 Layout * * * F_order;
```

**Strategy 5**: Align all the regions to 64 bytes while using the Fortran ordering of data

```
1 Task * GPU,CPU; # for any task, run on GPU if supported
2
3 Region * * GPU FBMEM; # for any task, any region, if mapped onto GPU, use FBMEM as default
4 Region * * CPU SYSMEM; # if mapped onto CPU, use SYSMEM as default
5
6 mcpu = Machine(CPU);
7 mgpu = Machine(GPU);
8
9 ========== Above is fixed ==========
10
11 Layout * * * Align==64 F_order;
```

**Strategy 6**: Place the task calculate_new_currents onto CPU

```
1 Task * GPU,CPU; # for any task, run on GPU if supported
2
3 Region * * GPU FBMEM; # for any task, any region, if mapped onto GPU, use FBMEM as default
4 Region * * CPU SYSMEM; # if mapped onto CPU, use SYSMEM as default
5
6 mcpu = Machine(CPU);
7
8 mgpu = Machine(GPU);
9
10 Layout * * * SOA C_order;
11
12 ========== Above is fixed ==========
13 Task calculate_new_currents CPU;
```

**Strategy 7**: Collect all the memory used by task calculate_new_currents

```
1 Task * GPU,CPU; # for any task, run on GPU if supported
2
3 Region * * GPU FBMEM; # for any task, any region, if mapped onto GPU, use FBMEM as default
4 Region * * CPU SYSMEM; # if mapped onto CPU, use SYSMEM as default
5
6 mcpu = Machine(CPU);
7 mgpu = Machine(GPU);
8
9 Layout * * * SOA C_order;
10
11 ========== Above is fixed ==========
12 CollectMemory calculate_new_currents *;
```

**Strategy 8**: Ensure that at most 4 tasks of calculate_new_currents can be run at the same time

```
1 Task * GPU,CPU; # for any task, run on GPU if supported
2
3 Region * * GPU FBMEM; # for any task, any region, if mapped onto GPU, use FBMEM as default
4 Region * * CPU SYSMEM; # if mapped onto CPU, use SYSMEM as default
5
6 mcpu = Machine(CPU);
7 mgpu = Machine(GPU);
8
9 Layout * * * SOA C_order;
10
11 ========== Above is fixed ==========
12 InstanceLimit calculate_new_currents 4;
```

**Strategy 9**: Map the second region argument of task distribute_charge onto GPU's Zero-Copy memory

```
1 Task * GPU,CPU; # for any task, run on GPU if supported
2
3 Region * * GPU FBMEM; # for any task, any region, if mapped onto GPU, use FBMEM as default
4 Region * * CPU SYSMEM; # if mapped onto CPU, use SYSMEM as default
5
6 mcpu = Machine(CPU);
7 mgpu = Machine(GPU);
8
9 Layout * * * SOA C_order;
10
11 ========== Above is fixed ==========
12 Region distribute_charge 1 GPU ZCMEM;
```

**Strategy 10**: Map the tasks of calculate_new_currents,distribute_charge,update_voltages onto GPUs in a 1D cyclic manner: perform a cyclic distribution over both the node and processor dimensions.

```
1  Task * GPU,CPU; # for any task, run on GPU if supported
2
3  Region * * GPU FBMEM; # for any task, any region, if mapped onto GPU, use FBMEM as default
4  Region * * CPU SYSMEM; # if mapped onto CPU, use SYSMEM as default
5
6  mcpu = Machine(CPU);
7  mgpu = Machine(GPU);
8
9  Layout * * * SOA C_order;
10
11 ========== Above is fixed ==========
12 def cyclic1d(Task task) {
13     ip = task.ipoint;
14     # cyclic over node, cyclic over gpu
15     return mgpu[ip[0] % mgpu.size[0], ip[0] / mgpu.size[0] % mgpu.size[1]];
16 }
17
18 IndexTaskMap calculate_new_currents,distribute_charge,update_voltages cyclic1d;
```

### A.8. Generated Mapper Examples

Here we provide examples of generated mappers for a subset of problems. The mappers, written in DSL, are produced by the mapper agent. While the LLM is responsible for creating and refining the mapper agent, the agent itself is implemented in Python, and it generates mappers as DSL programs. For the Circuit Simulation benchmark, the optimized mapper (Figure A5) is more concise than the initial version (Figure A4), with an additional constraint for byte alignment in the data layout. In contrast, for Solomonik's algorithm, the initial mapper is relatively simple (Figure A6), whereas the final optimized mapper adopts a more complex and detailed index mapping strategy (Figure A7).

```
1 Task * GPU,OMP,CPU;
2 Task calculate_new_currents GPU;
3 Task update_voltages GPU;
4 Region * * GPU FBMEM;
5 Region * * * SOCKMEM,SYSMEM;
6 Region * all_times GPU FBMEM;
7 Region * all_nodes GPU FBMEM;
8 Region * all_wires GPU FBMEM;
9 Region * ghost_ranges GPU FBMEM;
10 Region * rp_all_nodes GPU FBMEM;
11 Region * all_private GPU FBMEM;
12 Region * all_shared GPU FBMEM;
13 Region * rp_shared GPU FBMEM;
14 Region * rp_wires GPU FBMEM;
15 Region * rp_ghost_ranges GPU FBMEM;
16 Layout * * * C_order AOS;
17 mgpu = Machine(GPU);
18
19 m_2d = Machine(GPU);
20 def same_point(Task task) {
21     return m_2d[*task.parent.processor(m_2d)];
22 }
```

*Figure A4.* For the Circuit task, we show the mapper produced by the mapper agent at iteration 2.

```
1 Task * GPU,OMP,CPU;
2 Task calculate_new_currents GPU;
3 Task update_voltages GPU;
4 Region * * GPU FBMEM;
5 Layout * * * C_order AOS Align==128;
6 mgpu = Machine(GPU);
7
8 m_2d = Machine(GPU);
9 def same_point(Task task) {
10     return m_2d[*task.parent.processor(m_2d)];
11 }
```

*Figure A5.* For the Circuit task, we show the mapper produced by the mapper agent at iteration 10.

```
1  Task * GPU,OMP,CPU;
2  Region * * GPU FBMEM;
3  Region * * * SOCKMEM,SYSMEM;
4  Layout * * * F_order SOA;
5  mgpu = Machine(GPU);
6
7  def block1d(Task task) {
8      ip = task.ipoint;
9      return mgpu[ip[0] % mgpu.size[0], ip[0] % mgpu.size[1]];
10 }
11
12 IndexTaskMap task_2 block1d;
13
14 m_2d = Machine(GPU);
15 def same_point(Task task) {
16     return m_2d[*task.parent.processor(m_2d)];
17 }
```

*Figure A6.* For Solomonik's algorithm, we show the mapper produced by the mapper agent at iteration 2.

```
1  Task * GPU,OMP,CPU;
2  Region * * GPU FBMEM;
3  Region * * * SOCKMEM,SYSMEM;
4  Layout * * * C_order SOA No_Align;
5  mgpu = Machine(GPU);
6
7  def linearize3D(Task task) {
8      ip = task.ipoint;
9      linearize = ip[0] + ip[1] + ip[2];
10     return mgpu[linearize % mgpu.size[0], linearize % mgpu.size[1]];
11 }
12
13 IndexTaskMap task_1 linearize3D;
14
15 def linearize2D(Task task) {
16     ip = task.ipoint;
17     linearize = ip[0] * 2 + ip[2];
18     return mgpu[linearize % mgpu.size[0], linearize % mgpu.size[1]];
19 }
20
21 IndexTaskMap task_1 linearize2D;
22 IndexTaskMap task_2 linearize2D;
23 IndexTaskMap task_3 linearize2D;
24 IndexTaskMap task_5 linearize2D;
25
26 m_2d = Machine(GPU);
27 def same_point(Task task) {
28     return m_2d[*task.parent.processor(m_2d)];
29 }
```

*Figure A7.* For Solomonik's algorithm, we show the mapper produced by the mapper agent at iteration 10.

