# OpenReview forum: "Improving Parallel Program Performance with LLM Optimizers via Agent-System Interfaces"
_ICML.cc/2025/Conference — ICML 2025 poster_

### Official Review · Reviewer_edWw · 2025-03-13

**Overall Recommendation:** 3

**Summary:**

The paper proposes a system for automatically generating and optimizing parallel program mappers. Particularly, it tries to do this via using  a generative optimization approach aided by an "agent-system interface" which uses a DSL to allow LLMs to write code at a high-level.
Empirical results show that it can sometimes even beat expert-written mappers.

**Claims And Evidence:**

The authors claim that Agent-System Interface (an abstraction layer between the agent and the system) simplifies code generation and provides more meaningful feedback to the agent. This includes the DSL design and AutoGuide mechanisms. Both of these are supported by evidence from Sections 5.2 and 5.3. Performance results (against OpenTuner and Human Baselines) show strong performance of the system.

**Essential References Not Discussed:**

To the best of my knowledge, relevant works are discussed appropriately.

**Experimental Designs Or Analyses:**

The paper uses various relevant baselines in the experiments. The ablations are also well directed to properly evaluate individual contributions of the ASI block, providing evidence for improvements from both DSL design and AutoGuide.

**Methods And Evaluation Criteria:**

The proposed methodology is well-motivated and explained clearly. The ASI design with customized DSL and explain-suggest based feedback is novel within the domain context and likely helpful for future work. For evaluation, the paper uses 9 HPC benchmarks and measures speedup achieved by the proposed approach.

**Other Comments Or Suggestions:**

The authors can motivate the choice of benchmarks better, particularly for general readers unfamiliar with domain knowledge of these tasks.

**Other Strengths And Weaknesses:**

Strengths:
- Comprehensive experiments on standard HPC benchmarks, with clear baselines and ablations.
- Agent-System Design with proper ablations

Weaknesses
- Since only 9 benchmarks are considered, it is not clear how to estimate the generality of the approach
- It is possible the DSL and relevant prompts bake in a lot of human prior which can lead to inflated estimation of underlying model capabilities. Authors should also attempt to study sensitivity to the design of this ASI

**Questions For Authors:**

How much domain expertise (about the benchmark domain and the LLM capabilities) is needed to expand the ASI and the prompt for the benchmark? More practically, do we need experts who write the mappers to design DSL encoding appropriate priors?

**Relation To Broader Scientific Literature:**

The paper builds upon prior HPC / autotuning literature and promotes generative optimization as an alternative to prior reinforcement learning approaches.

**Theoretical Claims:**

none

---

> ### Author Rebuttal · Authors · 2025-04-01
>
> Thank you for the thoughtful and constructive review.
>
> **Q1: How much domain expertise (about the benchmark domain and the LLM capabilities) is needed to expand the ASI and the prompt for the benchmark? More practically, do we need experts who write the mappers to design DSL encoding appropriate priors?**
>
>
> Our framework is designed to minimize the need for domain or system-level expertise when adding new benchmarks. Specifically, for all benchmarks in our evaluation, the agent begins from a randomly initialized mapper and improves it through generative optimization, without any handcrafted priors or expert-written strategies.
> To incorporate a new application into the benchmark, only two inputs are needed:
> 1) **Application metadata**, including task names and the list of data arguments each task accesses
>
> 2) **Hardware specification**, including the number of CPUs and GPUs per node and the total number of nodes.
>
>
> These inputs are typically available from the application code and the machine, requiring no understanding of the DSL or the runtime system. As a result, application developers do not need to write or understand mappers, provide mapping hints, or possess any knowledge of the DSL.
>
>
> That said, our system can also **support optional injection of expert knowledge** via the `AutoGuide` module. This module can provide customized interpretations of execution failures or application-specific heuristics (e.g., mapping large tasks to GPUs if the developer already knows which tasks are large). While our experiments do not use any such guidance, we believe this optional flexibility is valuable in practice, especially for developers who already have insight into performance bottlenecks and want to speed up the optimization process.
>
> **Q2: The authors can motivate the choice of benchmarks better, particularly for general readers unfamiliar with the domain knowledge of these tasks.**
>
> Thanks for the suggestion! We will improve the paper to provide more context and motivation for our benchmark selection. Among the 9 benchmarks, 6 are well-known matrix multiplication algorithms (Cannon, SUMMA, PUMMA, Johnson’s, Solomonik’s, and COSMA). Parallel matrix multiplication remains an active research topic due to its central role in high-performance computing and scientific simulations [1]. Furthermore, improving matrix multiplication performance has a broad impact, as it accelerates numerous downstream machine learning workloads [2,3]. The remaining 3 applications (Circuit, Stencil, and Pennant) represent diverse scientific computing workloads beyond matrix multiplication. Together, this benchmark suite offers both depth (through representative matrix-multiplication algorithms) and breadth (through diverse HPC workloads). We will expand the background and motivation accordingly in the revision.
>
>
> **References:**
>
> [1] Yadav et al. “DISTAL: The Distributed Tensor Algebra Compiler”. 2022
>
> [2] Jangda et al. “Breaking the Computation and Communication Abstraction Barrier in Distributed Machine Learning Workloads”. 2022
>
> [3] Zheng et al. “TileLink: Generating Efficient Compute-Communication Overlapping Kernels using Tile-Centric Primitives”. 2025

---

### Official Review · Reviewer_vUWr · 2025-03-13

**Overall Recommendation:** 3

**Summary:**

This paper proposes a system powered by large language models (LLMs) to automate both the generation and optimization of mapper code. Specifically, it introduces a Domain-Specific Language (DSL) that provides a high-level interface encapsulating all performance-critical decisions required for mapper generation. The authors then implement the AutoGuide mechanism, which interprets raw execution outputs into informative and actionable feedback. This mechanism enables the agent to iteratively optimize the mapper by leveraging enriched feedback to refine its code generation strategy. Finally, the authors apply generative optimization to further enhance the generated code.
Evaluation results demonstrate that the proposed method outperforms OpenTuner even after 1,000 iterations, achieving a 3.8× performance improvement.

**Claims And Evidence:**

1. This paper argues that the proposed DSL simplifies mapper code generation and provides valuable guidance to the agent. The results demonstrate that the DSL requires less code and enhances the optimization process.

2. The paper claims that the agentic framework achieves up to a 1.34× speedup across nine benchmarks, outperforming expert-written mappers while reducing tuning time from days to minutes. As shown in Fig. 4, the performance gains are indeed significant.

**Essential References Not Discussed:**

NA

**Experimental Designs Or Analyses:**

In the ablation study, the authors present Code Generation Success Rates. However, in the overall evaluation, only the performance results are reported. Additionally, I am curious about the Random Mapper— is it possible that all these random mappers are correct?

**Methods And Evaluation Criteria:**

1. One key contribution of this work is the proposed DSL. However, the authors do not describe its grammar; instead, they illustrate it with a single example. Furthermore, it remains unclear whether the proposed DSL is expressive enough to cover all mapper code generation problems.  Another unclear part is that the authors mention they use server specifications and application information as inputs but did not provide more details.

2. Regarding the evaluation (Fig. 4), why does the paper report only the best results of the proposed method rather than presenting the full optimization curve?

**Other Comments Or Suggestions:**

NA

**Other Strengths And Weaknesses:**

1. This paper is well-written and easy to follow.

2. The studied problem is interesting and important.

**Questions For Authors:**

1. What is your DSL grammar.

2. Why only report the best point for the proposed appraoch?

**Relation To Broader Scientific Literature:**

It is important for improving system performance.

**Theoretical Claims:**

NA

---

> ### Author Rebuttal · Authors · 2025-04-01
>
> Thank you for the thoughtful and constructive feedback.
>
> **Q1: What is your DSL grammar?**
>
> In the revision, we will include a complete description of the DSL syntax in the Appendix, covering its constructs for task placement, memory allocation, layout specification, and index mapping, as shown below.
>
>
> ```
> Terminals:
>   TaskName, RegionName, var, int
>
> Grammar Rules:
>   Program         → Statement+
>   Statement       → TaskMap | DataMap | DataLayout | FuncDef | IndexTaskMap TaskName var
>   TaskMap         → Task TaskName Proc+
>   DataMap         → Region TaskName RegionName Proc Memory+
>   Proc            → CPU | GPU | OMP
>   Memory          → SYSMEM | FBMEM | ZCMEM | SOCKMEM
>   DataLayout      → Layout TaskName RegionName Proc Constraint+
>   Constraint      → SOA | AOS | C_order | F_order | Align == int
>   FuncDef         → def var(var+): FuncStmt+
>   FuncStmt        → var = Expr | return Expr
>   Expr            → var | var(Expr+) | Machine(Proc) | Expr.Expr | Expr Op Expr |
>                     (Expr) | Expr[Expr] | *Expr | Expr ? Expr : Expr
> ```
>
>
> **Q2: Why only report the best point for the proposed approach rather than full optimization curve?**
>
> Please kindly note that we report both the best and average optimization trajectory over 10 iterations across 5 runs in Figure 4. Also, we want to clarify that reporting the best-performing mapper is appropriate in our context. Mapper optimization is an offline process, and in practice, it is standard to run the optimizer multiple times and deploy the best result. Once identified, the mapper can be reused across repeated executions on the same application, input, and hardware, incurring no further search cost.
>
> **Q3: Is the DSL expressive enough to cover all mapper code generation problems?**
>
> Our DSL is designed to express a wide range of high-performance mapping strategies, including all of the most important decisions. While there may be cases where certain optimizations are not directly expressible, we have not encountered any. Despite being more constrained than general-purpose C++, the DSL has been proven to be effective: all mappers discovered by our agent that outperform expert-written C++ implementations are expressible within the current DSL.
>
> **Q4: Server specifications and application information are mentioned as inputs, but details are missing.**
>
> We will clarify these details in the revised version. The server specification includes the number of GPUs and CPUs, and whether the OpenMP runtime is enabled. The application specification includes the list of tasks defined in the application and the data regions accessed by each task. These inputs define the structured search space explored by the agent during optimization. We will also include an example of such input specifications in the revised version for completeness.
>
>
> **Q5: The authors present Code Generation Success Rates (Section 5.2). However, in the main evaluation (Section 5.1), only the performance results are reported.**
>
>
> In the main evaluation (Section 5.1), we focus on measuring end-to-end performance, which includes both the correctness and performance of generated mappers. If the generated code has any syntax or runtime issues, its throughput is recorded as 0. Therefore, the performance numbers in Section 5.1 implicitly reflect code generation success, i.e., incorrect mappers yield zero performance.
>
>
> Section 5.2 isolates the code generation aspect to better analyze the effects of the DSL on LLM generation success. This section does not aim to evaluate performance directly but rather investigates how often the LLM produces syntactically and semantically correct mappers given natural language descriptions. It complements the main results by demonstrating that using the DSL significantly improves generation success compared to C++, which underpins the performance improvements seen in Section 5.1.
>
> **Q6: Are all random mappers correct?**
>
> No, not all random mappers are correct. For each application, we generate 10 random mappers by sampling from the full DSL-defined search space, totaling 90 mappers across 9 applications. Among them, 74 (82.2%) raise runtime errors due to invalid mapping decisions. The runtime system enforces correctness by rejecting such mappers during execution, resulting in a throughput of zero.

---

### Official Review · Reviewer_kSxf · 2025-03-14

**Overall Recommendation:** 3

**Summary:**

The paper introduces an innovative framework aimed at automating the process of optimizing parallel program performance using large language models. The proposed system employs a Domain-Specific Language to simplify the generation of mapping code and uses a mechanism called AutoGuide to turn raw execution feedback into actionable insights for the optimization agent. This system leverages generative optimization to find high-performance mappers efficiently, achieving superior results compared to existing tools like OpenTuner, even after fewer iterations.

**Claims And Evidence:**

1. The experiments are conducted on a single-node setup with specific hardware. Whether the approach can achieve similar results on larger, more heterogeneous systems with diverse hardware remains questionable.

2. Average performance across multiple runs is reported, but reports on other percentile of performance are missing. This makes it hard to judge the consistency and reliability of the speedup. Additionally, how is the tail performance being impacted?

**Essential References Not Discussed:**

N/A

**Experimental Designs Or Analyses:**

The authors evaluate their approach using nine established benchmarks from the Legion framework, which is appropriate for assessing parallel program performance. A few potential issues:

1. Experiments on a single-node hardware design
2. Performance measured only on average is not sufficient, tail performance should also be reported

**Methods And Evaluation Criteria:**

Yes

**Other Comments Or Suggestions:**

N/A

**Other Strengths And Weaknesses:**

### Strengths

1. The system handles complex parallel computation systems and scales efficiently with larger programs and datasets.
2. The DSL abstracts away the complexities of low-level programming, making it easier for LLMs to generate correct mapping code and improving code generation success rates.

### Weaknesses

1. The current system works best for parallel programs that align with the DSL’s design. For non-standard or highly unique system architectures, the framework may need further customization.
2. While the AutoGuide mechanism is powerful, it still relies on raw execution output which might not always provide the necessary insights, especially in cases of complex or obscure errors.
3. Performance metrics could be extended.

**Questions For Authors:**

N/A

**Relation To Broader Scientific Literature:**

The paper advances parallel programming by synthesizing high-level DSL design, reinforcement learning autotuning, and LLM-driven code generation, incorporating natural language feedback and a modular, agent-based approach.

**Theoretical Claims:**

This paper focuses on experimental validation and does not include formal proofs for its theoretical claims.

---

> ### Author Rebuttal · Authors · 2025-04-01
>
> Thank you for your thoughtful review! We address the concerns below:
>
> **Q1: Performance metrics could be extended**
>
>
> We appreciate your suggestion and welcome the opportunity to clarify our evaluation methodology and expand the reported statistics.
>
> In our setting, reporting the best result across multiple runs is appropriate, as the best mapper is the one that is desired by the user. Mapper search is an offline optimization process, and it is feasible to run the optimizer multiple times to select the highest-performing mapper. Once identified, this mapper can be reused without incurring additional search cost, as the deployment scenario (application, input, and hardware) remains fixed.
>
> That said, we agree that additional statistics provide a more complete picture of performance variations. In the revised version, we will include the mean, standard deviation, worst, median, and best normalized throughput across five runs for each benchmark. The extended results are as follows:
>
> ### Our Framework (Normalized Throughput)
> | Benchmark  | Mean   | Std Dev | Worst  | Median | Best |
> |------------|--------|---------|--------|--------|------|
> | Circuit    | 1.33× | 0.01 | 1.31× | 1.33× | 1.34× |
> | Stencil    | 1.01× | 0.01 | 1.00× | 1.01× | 1.02× |
> | Pennant    | 1.03× | 0.02 | 1.00× | 1.03× | 1.04× |
> | Cannon     | 1.09× | 0.00 | 1.08× | 1.09× | 1.09× |
> | SUMMA      | 0.86× | 0.48 | 0.00× | 1.07× | 1.09× |
> | PUMMA      | 0.57× | 0.55 | 0.00× | 0.66× | 1.09× |
> | Johnson    | 0.98× | 0.17 | 0.68× | 1.06× | 1.07× |
> | Solomonik  | 0.52× | 0.41 | 0.00× | 0.61× | 1.09× |
> | COSMA      | 1.25× | 0.03 | 1.23× | 1.23× | 1.31× |
>
> We additionally report OpenTuner results.
>
> ### OpenTuner (Normalized Throughput)
> | Benchmark  | Mean   | Std Dev | Worst  | Median | Best |
> |------------|--------|---------|--------|--------|------|
> | Circuit    | 0.97× | 0.16 | 0.81× | 0.99× | 1.20× |
> | Stencil    | 0.00× | 0.00 | 0.00× | 0.00× | 0.00× |
> | Pennant    | 0.00× | 0.00 | 0.00× | 0.00× | 0.00× |
> | Cannon     | 0.00× | 0.00 | 0.00× | 0.00× | 0.00× |
> | SUMMA      | 0.00× | 0.00 | 0.00× | 0.00× | 0.00× |
> | PUMMA      | 0.00× | 0.00 | 0.00× | 0.00× | 0.00× |
> | Johnson    | 0.00× | 0.00 | 0.00× | 0.00× | 0.00× |
> | Solomonik  | 0.00× | 0.00 | 0.00× | 0.00× | 0.00× |
> | COSMA      | 0.00× | 0.00 | 0.00× | 0.00× | 0.00× |
>
> Our method achieves relatively stable performance across most benchmarks. The higher variance and occasional 0.00× worst-case throughput observed in SUMMA, PUMMA, and Solomonik are due to invalid mapper configurations in the search space (e.g., violating cuBLAS layout constraints). The runtime enforces correctness by rejecting such configurations during execution. While the generative optimizer typically learns to avoid these cases through the AutoGuide mechanism, occasional failures within the 10-iteration budget are still possible. In practice, such failures can be mitigated by repeating the optimization and selecting the best-performing mapper.
>
> In contrast, OpenTuner, despite running the same number of iterations, fails to generate valid mappers for 8 out of 9 benchmarks. This highlights the difficulty of exploring the search space using traditional reinforcement learning methods.
>
>
> **Q2: Experiments on multi-node systems**
>
> Thank you for raising this point. Our system supports multi-node execution, as this capability is already provided by the underlying runtime system. There is no fundamental technical limitation in our approach that prevents generalization to multi-node systems. The only technical issue we encountered in scaling up our experiments is an engineering issue, not a limitation of our method. Specifically, the cluster we currently use permits interactive execution on a single node, but multi-node runs require submitting jobs through the SLURM scheduler and waiting for resource allocation and job execution. While this can be addressed through SLURM-specific customization, it is orthogonal to the core contributions and novelty of our system. For users with direct multi-node access (e.g., without queueing), our system runs seamlessly without modification. We plan to extend our evaluation to larger-scale experiments as part of future work.

---

### Decision · Program_Chairs · 2025-05-01

**Decision:**

Accept (poster)

**Comment:**

This paper proposes an agent-based framework for automating mapper generation in parallel programs using LLMs, incorporating a custom DSL and execution-guided feedback (AutoGuide). The approach shows promising performance improvements across nine benchmarks, outperforming both OpenTuner and expert-written mappers within a small number of iterations.

All three reviewers recommend weak accept. They commend the novelty of the Agent-System Interface, clear experimental results, and practical utility for HPC. Key concerns include limited theoretical analysis, the scope of DSL expressiveness, and evaluation restricted to single-node hardware and a small benchmark suite. The authors’ rebuttal addressed most points thoroughly, adding statistical performance metrics, clarifying DSL design and search space coverage, and justifying the benchmark choices. Reviewers acknowledged these improvements and maintained their scores.

Overall, while some scalability and generalisation concerns remain, the paper makes a solid and novel systems contribution.